A user-embedded temporal attention neural network for IoT trajectories prediction

Feng Dongdong fengdd@chinatelecom.cn
Li Siyao
Xiang Yong
Zheng Jiahuan
China Telecom Research Institute , Guangzhou, Guangdong Province , China
Alatas Bilal
Electronic publication date: 2025 Feb 11
Publication date: 2025
Volume: 11
Electronic Location ID: e2681
Received 2024 Sep 19; Accepted 2025 Jan 14
Copyright: © 2025 Feng et al.
Copyright year: 2025
Copyright holder: Feng et al.
License: This is an open access article distributed under the terms of the Creative Commons Attribution License, which permits unrestricted use, distribution, reproduction and adaptation in any medium and for any purpose provided that it is properly attributed. For attribution, the original author(s), title, publication source (PeerJ Computer Science) and either DOI or URL of the article must be cited.
License URL: https://creativecommons.org/licenses/by/4.0/

Keywords: User embedding, Sequential recommendation, Attention mechanism, Trajectory prediction

Funding: China Telecommunication Research Institute This research is funded by the China Telecommunication Research Institute through a grant to the Mobile Research Institute of China Telecommunication Research Institute for the Research on the 5G network data analytics function (NWDAF) program. The funders had no role in study design, data collection and analysis, decision to publish, or preparation of the manuscript.

==============================
Over the past two decades, sequential recommendation systems have garnered significant research interest, driven by their potential applications in personalized product recommendations. In this article, we seek to explicitly model an algorithm based on Internet of Things (IoT) data to predict the next cell reached by the user equipment (UE). This algorithm exploits UE embedding and cell embedding combining the visit time interval information, and uses sliding window sampling to process more UE trajectory data. Furthermore, we use the attention mechanism, removed the query matrix operation and the attention mask, to obtain key information in data and reduce the number of parameters to speed up training. In the prediction layer, combining the positive and negative sampling and computing cross entropy loss also provides assistance to increase the precision and dependability of the entire model. We take the six adjacent cells of the current cell as candidates due to the limitation of the space problem, from which we predict the next destination cell of track movement. Extensive empirical study shows the recall of our algorithm reaches 0.5766, which infers the optimal result and high performance of our model.

Introduction

Motivation

In the age of the Internet of Things (IoT), an immense volume of data is produced by countless devices, surpassing the processing capabilities of traditional data processing services. To extract maximum value from this massive amount of data, time series analysis is essential. Time sequences and time series data are closely related, necessitating the development of advanced algorithms that can efficiently process and analyze these large datasets.

Problem statement

Sequential recommendation has gained significant research interest over the past two decades due to its potential applications in commodity recommendation. Traditional methods such as the Markov model, which relies on a state transition matrix for forecasting future behavior, face challenges in capturing intermittent transfers due to sparse sequence data. With the advent of deep learning, models based on recurrent neural networks (RNNs) and long short-term memory (LSTM) (Hochereiter & Schemidhuber, 1997; Gers, Schmidhuber & Cummins, 2000) networks have shown superior performance in handling sequential data. However, LSTM-based models face limitations in parallel computing, leading to the widespread adoption of attention mechanisms for sequence prediction.

Recent advancements in attention-based (Vaswani et al., 2017) models, such as self-attention based sequential model (SASRec) (Kang & McAuley, 2018), have demonstrated promising results in sequential recommendation. Despite these advancements, there remains a gap in effectively leveraging IoT data to predict user equipment (UE) trajectories in mobile networks.

Current trajectory prediction methods face several key challenges: Reliance on GPS data: Many models depend heavily on GPS coordinates, which can be challenging to obtain or may be inaccurate in certain environments.

Handling long sequences: Existing models often struggle to capture long-term dependencies and handle long sequences effectively.

Computational efficiency: Traditional models are computationally expensive and not well-suited for parallel processing, leading to slow training times.

Generalization and overfitting: Some models are prone to overfitting and do not generalize well to new data.

Contribution and novelty

To address these gaps, we propose a User-Embedded Temporal Attention Neural Network (UETANN) for predicting the next cell reached by UE in IoT environments. Our contributions are as follows: UE embedding: By incorporating UE embedding, we enable the model to learn the behavioral characteristics of users. This allows the model to predict locations based on similar user behaviors, effectively handling sparse sequence data and improving generalization capabilities.

Optimized attention mechanism: We enhance the attention mechanism by eliminating redundant query matrices and attention masks. This reduction in parameters not only speeds up training but also enhances computational efficiency, addressing a gap that previous studies have not adequately addressed.

Sliding window sampling: We utilize sliding window sampling to leverage more trajectory information. This approach enhances the model’s ability to capture complex patterns and long-term dependencies, which are often overlooked in traditional models.

Negative sampling: By selecting nearby locations as negative samples, we help the model better distinguish between positive and negative samples, thereby increasing prediction accuracy. This technique fills a gap in the literature where previous methods have struggled to improve precision in dense and overlapping spatial distributions.

Our extensive empirical study shows that the recall of our algorithm reaches 0.5766, indicating the optimal result and high performance of our model. This performance is particularly significant in the context of IoT trajectory prediction, where accurate and timely predictions are crucial for optimizing network resources and enhancing user experience.

Related works

Existing approaches for trajectory prediction

Several recent studies have explored the use of deep learning and attention mechanisms for trajectory prediction and recommendation. For instance, Spatial temporal recurrent neural networks (STRNN) (Liu et al., 2016) uses time and space interval information to improve location recommendation precision (Wu et al., 2016), while Spatio-temporal gated network (STGN) (Zhao et al., 2022) enhances LSTM models for better performance. Attention based spatiotemporal LSTM (ATST-LSTM) (Huang et al., 2021) and spatial-temporal attention networks (STAN) (Luo, Liu & Liu, 2021) also leverage spatial-temporal information to improve model accuracy.

Recent works, such as the Augmented Intelligence of Things for Emergency Vehicle Secure Trajectory Prediction and Task Offloading (Wu et al., 2024), TRACE: Transformer-based Continuous Tracking Framework Using IoT and MCS (Mohammed, Singh & Otrok, 2024), and Trajectory Simplification and Adaptive Map Matching Algorithm for Electric Bicycle (Wang, Liu & Yu, 2023), highlight the importance of integrating IoT data and advanced machine learning techniques for real-time tracking and prediction. These studies provide valuable insights into the challenges and opportunities in the field.

Sequential recommendation models

Markov model: The Markov model is one of the foundational approaches for sequential recommendations. It uses a state transition matrix to forecast future behavior. However, the Markov model struggles to capture intermittent transfer trips due to the sparsity of sequence data.

RNN-based models: Recurrent neural networks (RNNs) have been widely used for sequential prediction tasks, such as (Zhang et al., 2014). They form a strong baseline for many applications.

LSTM-based models: LSTM networks, an extension of RNNs, have shown improved performance in handling sequential data (Kong & Wu, 2018), as well as the gated recurrent unit (GRU) (Chung et al., 2014).

Metric embedding algorithms: These algorithms, such as those described in Feng et al. (2015), enhance the representation of items in the recommendation system.

Convolutional neural networks (CNN): CNNs have also been applied to sequential recommendation tasks, as detailed in Tang & Wang (2018).

Graph networks: Graph-based models, as described in Wu et al. (2019), leverage the relationships between items to improve recommendation accuracy.

Reinforcement learning algorithms: Reinforcement learning techniques, as outlined in Massimo & Ricci (2018), have been used to optimize sequential recommendations.

SASRec: SASRec is a notable model that uses attention mechanisms to improve sequential recommendation. Models developed based on SASRec have shown promising results.

Attention mechanisms in IoT

ATST-LSTM: This model combines attention mechanisms with LSTM to improve the performance of trajectory prediction.

STAN: STAN also utilize attention mechanisms to enhance the accuracy of trajectory prediction.

GETNext: This algorithm recommends POIs by constructing a trajectory flow map between points of interest (POIs) and embedding POIs through graph convolutional networks (GCNs) (Yang, Liu & Zhao, 2022).

Trajectory prediction without GPS data

Point-of-Interest recommendation algorithm combining temporal features and collaborative filtering: This algorithm uses temporal features and collaborative filtering to improve location recommendation precision but does not consider time interval information, leading to suboptimal performance (Song et al., 2016).

MEMO: Proposed by Huang et al. (2022) MEMO effectively utilizes multiple network representations to learn heterogeneous relationships and explicitly combines temporal user POI interactions with coupled recurrent neural networks.

SASRec: As mentioned, SASRec uses attention mechanisms to capture the dependencies in sequential data, making it suitable for various IoT applications.

Limitations of current methods

Despite their effectiveness in certain scenarios, existing methods have several limitations: Reliance on GPS data: Many models depend heavily on GPS coordinates, which can be challenging to obtain or may be inaccurate in certain environments.

Handling long sequences: Existing models often struggle to capture long-term dependencies and handle long sequences effectively.

Computational efficiency: Computing efficiency is an important consideration for IoT (Potluri et al., 2011; Feng et al., 2015), while traditional models are computationally expensive and not well-suited for parallel processing, leading to slow training times.

Generalization and overfitting: Some models are prone to overfitting and do not generalize well to new data.

Comparison with existing methods

Our proposed model overcomes the limitations of existing methods in the following ways: Independence from GPS data: Our model does not rely on GPS data, making it suitable for scenarios where location coordinates are missing or inaccurate.

Handling long sequences: By using sliding window sampling, our model leverages more sequence information, improving its ability to handle long sequences effectively.

Reducing computational load: Optimizing the attention mechanism by removing redundant query matrices and attention masks reduces the number of parameters, speeding up the training process.

Enhancing generalization: Sliding window sampling and negative sampling techniques help the model generalize better, reducing overfitting and improving prediction accuracy.

Preliminaries

This part gives the question formulation and definition of terms. We represent the sets of UE, cell and timestamp as U={u1,u2,…,uU},C={c1,c2,…,cC},T={t1,t2,…,tT}, respectively. Figure 1 shows how the trajectories look like, unlike GPS data, our UEs do motions between cells.

Figure 1 Cell trajectory.

History trajectory

A UE trajectory is temporally ordered arrivals. Each arrival ak of the trajectory of UE ui is a tuple ( ui, ck, tk), in which ck is the cell and tk is the timestamp. Each UE has a variable length trajectory tra(ui)={a1,a2,…,ami}, where mi is the total length of the ui trajectory. Every trajectory is transformed into several fixed length sequences seq(ui)={a1,a2,…,an}, with n as the maximum length of a single seq(ui) we consider. We only consider trajectories whose length is not less than 5. If mi < n, zeros are padded for cells to the left until the trajectory length is n, and we pad timestamp with the first timestamp ts1 where s1 is not a padded cell so that those time intervals will be zero after calculation as we can see in the “Time Intervals Embedding Layer” section. Samples of ui will be like this: from

(1) {a1,a2,...,an}topredictcn+1,

{a2,a3,...,an+1}topredictcn+2,⋯

{ami−n,ami−n+1,...,ami−1} topredictcmi.

We mask off the padding arrivals during computation.

Mobility prediction

Given the UE trajectory (a1,a2,…,am), our target is to predict the next arriving cell c ∈ am+1.

Proposed framework

In this section, we present a detailed framework for predicting the next cell reached by UE using IoT data. Our method consists of four main components: 1) a multimodal embedding layer that learns representations of the UE, cell, and time intervals; 2) an attention layer that captures the relationships between arrivals within the UE trajectory; 3) a negative sampling technique that enhances the model’s ability to distinguish between positive and negative samples; and 4) a prediction layer that computes the scores of candidate cells. Additionally, we introduce a detailed flow diagram to visualize the model’s architecture.

The architecture of the proposed method is shown in Fig. 2.

Figure 2 The architecture of the proposed model.

Multimodal embedding layer

The multimodal embedding layer is designed to encode the UE, cell, and time intervals into latent representations. This layer comprises three sub-layers: the UE trajectory embedding layer, the positional embedding layer, and the time intervals embedding layer.

UE trajectory embedding layer

In order to encode UE, cell into latent representations, a multi-modal embedding layer is employed. We define the embedded representations of UE and cells as eu∈Rd and ec∈Rd, respectively, where d represents the latent dimension. The other modules use the embedding module to turn the scalars into dense vectors, accelerating representation and minimizing processing. We can find UEs with similar behavior through UE embedding.

For UEs with similar historical trajectories, there is a high probability that the cells arriving at the next time will overlap. The purpose of adding UE embedding is similar to that of collaborative filtering. The embedding vectors of similar UEs are also close in high-dimensional space.

Similar to UE embedding, we can find adjacent cells through cell embedding. The embedding vectors of adjacent cells are relatively close in high-dimensional space, making up for the lack of longitude and latitude of IoT cell data.

We embed both eu and ec into d dimension. Concretely, taking UE embedding for example, we create an embedding matrix of nu×d, where nu represents the number of UEs. A single UE is retrieved from this embedding matrix to obtain eu. ec is obtained through a similar approach. The output of the UE trajectory embedding layer for each arrival ea is the sum ea = eu + ec ∈Rd. We stack embeddings of arrivals together and result in a matrix MA∈Rn×d:

(2) MA=[ea1ea2…ean].

Positional embedding layer

Due to the difference between the attention mechanism and RNN, every cell embedding data is input at the same time with no priority. Thus, we use positional embedding to reflect the order of cell visits in the trajectory. For the self-attention mechanism’s keys and values, we employ two learnable positional embedding matrices MKP∈Rn×dandMVP∈Rn×d. After retrieval, we get the matrices EKP∈Rn×dandEVP∈Rn×d:

(3) EKP=[p1kp2k…pnk]EVP=[p1vp2v…pnv].

Time intervals embedding layer

We represent the relationship between two cells as the time period in a trajectory. Fewer UEs tend to be mobile more frequently than others. We are solely interested in a UE sequence’s relative time interval lengths. We take the shortest time interval of each sequence (apart from 0) in the UE sequence as the divisor to obtain the customized interval. The time interval diju = ⌊|ti−tj|dminu⌋, where ti, tj are timestamps of an n length time series t = ( t1, t2, . . . , tn), dminu is the shortest time interval (except for 0). Consequently, the UE relation matrix Ru∈Nn×n is:

(4) Ru=[d11ud21u…dn1ud12ud22u…dn2u…………d1nud2nu…dnnu].

This research limits the maximum relative time interval between two cells to l. We assume that after exceeding a certain threshold, the precise relative time interval is useless. Cutting the maximum interval can also avoid sparse relation encoding. Therefore, the clipping matrix is Rclippedu = clip ( Ru), where the clipping is applicable to each element diju=min(l,diju).

Similar to the positional embedding, we employ two learnable embedding matrices MKD∈Rl×dandMVD∈Rl×d to Rclippedu, then we get keys EKD ∈Rn×n×d and values EVD ∈Rn×n×d after retrieval:

(5) EKD=[d11kd21k…dn1kd12kd22k…dn2k…………d1nkd2nk…dnnk]

(6) EVD=[d11vd21v…dn1vd12vd22v…dn2v…………d1nvd2nv…dnnv].

Obviously, the two matrices are symmetric.

Attention layer

The attention layer is responsible for learning the relationships between the arrivals within the UE trajectory. By focusing on relevant parts of the trajectory, the attention mechanism helps the model capture long-term dependencies and improve prediction accuracy. We remove the query matrix operation and the attention mask to reduce the number of parameters and speed up training. The attention mechanism updates the representation of each arrival by weighting the contributions of other arrivals in the trajectory.

Self-attention layer

For each input sequence EA = ( ea1, ea2, . . ., ean) of n arrivals where eai ∈ Rd, compute a new sequence S = ( s1, s2, . . ., sn), where si ∈ Rd.

A weighted average of position/relation embeddings and linearly transformed input elements is calculated for each output element si:

(7) si=∑j=1n⁡αij(eajWV+pjv+dijv)

where WV ∈Rd×d is input projection for value.

Using a softmax function, each weight coefficient αij is calculated:

(8) αij=exp⁡(gij)∑k=1n⁡exp⁡(gik).

Using a function that takes inputs, relationships, and positions into account, gij is calculated as shown in Eq. (9):

(9) gij=eai(eajWK+pjk+dijk)Td

where WK ∈Rd×d is input projection for a key.

Note that unlike common processing method we do not use the query matrix WQ here, because eaiWQ(eajWK+pjk+dijk)T can be changed into the form of eai(eajWK+pjk+dijk)T, the steps are as shown in Eq. (10):

(10) eaiWQ(eajWK+pjk+dijk)T→eai[(eajWK+pjk+dijk)(WQ)T]T→eai[eajWK(WQ)T+pjk(WQ)T+dijk(WQ)T]T.

We take WK(WQ)T as new WK. Because pjk,dijk are the vectors obtained by embedding, so (WQ)T among pjk(WQ)Tanddijk(WQ)T can be further multiplied into the corresponding embedding matrix which is also to be learned. So far, the transformation has been completed. By removing the query matrix, we can reduce the number of parameters and speed up training. Of course, it is possible to remove the query matrix through this transformation in this article, but it may not be feasible in other algorithms, such as STAN (Luo, Liu & Liu, 2021). When the dimension is big, the scale factor d is employed to prevent the inner product from having large values.

Another note is that we do not use the attention mask, because we use sliding windows and only need the last information vector (as will be seen in the prediction layer) when learning sequence information, there is no problem of information leakage, so there is no need for an attention mask.

Forward network

Even though our distinctive attention layer can combine all prior arrivals, absolute position, and time intervals, they are combined linearly. Therefore, we use a Relu function to apply two linear transformations, which can endow the model with non-linearity:

(11) F(si)=Relu(siW1+b1)W2+b2

(12) Relu(x)=max(0,x)

where W1, W2∈Rd×d and b1, b2 ∈Rd.

As discussed in Kang & McAuley (2018), extra issues, such as over fitting, an unstable training process (e.g., disappearing gradients), the need for more training time, will appear after the attention layer and forward network are superimposed. Like Kang & McAuley (2018) and Vaswani et al. (2017), to address these issues, we additionally use layer normalization, residual connection, and dropout regularization techniques. Due to the random discarding of neurons during the training process, dropout regularization can prevent the model from relying too much on certain local features. Thus, it plays a regularization role in training the model and can be used to address overfitting issues.

(13) Si=si+Dropout(F(LN(si))).

Inputs are normalized across features using layer normalization. This can expedite the training of neural networks, according to Ba, Kiros & Hinton (2016). F is defined by Formula (10). Layer normalization is described as:

(14) LN(x)=α⊙x−μσ2+ϵ+β

where x is a vector including all features but not samples, μ and σ are the mean and variance, α and β are parameters to be learned, ϵ is a very small value to prevent the denominator from being zero.

Negative sampling

Negative sampling is used to transform the original prediction problem into a binary classification problem. For each positive sample (the actual next cell), we select several negative samples (nearby cells that are not the next cell). This approach helps the model better distinguish between positive and negative samples, thereby increasing prediction accuracy.

Recall that we convert UE trajectory to several fixed length sequences seq(u)={a1,a2,...,an}, ai=(u,ci,ti). We define on as the expected output given a n length UE sequence. Usually, we adopt negative sampling to increase score of positive cells and decrease score of negative cells. We use the six adjacent cells of cn as candidate cells, including cn−1 and the next cell that UE actually arrives, sometimes the two will coincide, the next cell that UE actually arrives is a positive sample, and we randomly select one of the remaining five cells as a negative sample. Thus, we get expected positive cell on and one negative cell on′ to obtain a set of pairwise samples D={(seq(u),o,o′)}.

Prediction layer

The prediction layer computes the score of candidate cells. Given the updated representations from the attention layer, the prediction layer calculates the likelihood of each candidate cell being the next destination. We use the six adjacent cells of the current cell as candidates due to spatial limitations. The scores are computed using a scoring function, and the cell with the highest score is selected as the predicted next cell.

We obtain the composite representation of arrivals, locations, and time intervals following stacked attention blocks. In the last block, we only compute sn, as well as αnj, gnj, Sn, regardless of previous n − 1 results. To distinguish from the previous expression, we rename the last output Sn to Sn′. To predict the next cell, we multiply the representation learned before and cell embedding to calculate UE’s preference score for cell i, as shown below:

(15) Ri,n=Sn′Ci

where Ci ∈Rd is the embedding of cell i and Sn′ is the representation given the previous n arrivals (i.e., as1, as2, . . ., asn) and their time intervals matrix Ru.

Cross-entropy loss

To optimize the model, we use cross-entropy loss. The loss function increases the score of positive samples and decreases the score of negative samples. This ensures that the model learns to accurately predict the next cell while maintaining robustness against incorrect predictions.

We use binary cross entropy as the loss function and a sigmoid function σ(x) = 1/(1 + exp(−x)) to make cell scores between zero and one:

(16) −∑D⁡[log⁡(σ(Ron,n))+log⁡(1−σ(Ron′,n))]+λ∥Θ∥F2

where ∥⋅∥F stands for the Frobenius norm, λ is the regularization parameter, and Θ is the set of embedding matrices.

The Adam (Kingma & Ba, 2017) optimizer is used to improve our suggested model. We use mini-batch SGD (Stochastic Gradient Descent) to increase training efficiency since the number of samples could be very big.

Detailed flow diagram

Figure 3 is a detailed flow diagram of the model architecture:

Figure 3 A detailed flow diagram of the model architecture.

Explanation of the flow diagram:

1. Input layer: Raw data (UE ID, cell ID, and check-in time) is fed into the model.

2. Multimodal embedding layer: UE trajectory embedding: Converts UE and cell IDs into dense vectors.

Positional embedding: Adds positional information to maintain sequence order.

Time intervals embedding: Encodes time intervals between arrivals.

3. Attention layer: Captures relationships between arrivals using a modified attention mechanism.

4. Negative sampling: Transforms the problem into a binary classification task by selecting negative samples.

5. Prediction layer: Candidate cell scoring: Computes scores for candidate cells.

Prediction: Selects the cell with the highest score as the next cell.

6. Loss function: Uses cross-entropy loss to optimize the model.

Experiment

Dataset

Data source

The experimental data are from the historical data detail record (DDR) generated by the IoT devices. We make experiments on two product forms, shared bicycle and goods locate. The time span is from June 20, 2022 to July 21, 2022. Figure 4 shows what UE trajectories may look like; some cells are frequently reached while others are not.

Figure 4 Heatmap of UE trajectories.

The above are private datasets, and we are also conducting experiments on public datasets, mainly using the following public datasets: MovieLens: The MovieLens dataset is a public dataset produced by the GroupLens project team. The MoveieLens dataset can be said to be one of the most classic datasets in the field of recommendation systems, which records when each user reviews what movies. We use the version MovieLens-1m, which contains over 1 million records (Harper & Konstan, 2015).

Gowalla: A check-in dataset, widely used for location prediction and POI prediction, which is introduced in Cho, Myers & Leskovec (2011).

Foursquare: Foursquare is a mobile service website based on user geographic location information, encouraging mobile users to share their current geographic location and other information with other people. The Foursquare dataset records this information, and is introduced in Sarwat et al. (2014).

Amazon: The Amazon dataset records users’ evaluations of Amazon website products and is a classic dataset for recommendation systems. We mainly selected two types, “Beauty” and “Electronics”, which are introduced in Ni, Li & McAule (2019).

Variable description

Since the private data is transmitted through a unified interface, the data has a standard format, including: Product information: industry categories, industry sub-categories, product forms, etc.;

Location information: cell ID, area code, etc.;

Device information: imsi, imei, etc.;

Communication information: Access point name (APN) identification, network access identifier (NAI) type, IP address, number of bytes sent/received, service type, roaming type, etc.;

Time information: start time, end time, duration, etc.

Data processing

The original private data is somewhat redundant. We only choose the records when UE enters a cell different from the current cell. Three fields are selected for further study: UE ID (imsi), cell ID and check in time (start time). At present, our algorithm only considers these three features, and adding more features such as the number of bytes sent and received may improve prediction accuracy. This may be a potential area for improvement in our algorithm in the future.

The UE and cell are sorted by their ID respectively, and then mapped to int to reduce memory requirements and accelerate training. The time interval information is processed according to the method in the “Time Intervals Embedding Layer” section before model training.

The basic dataset (including private and public dataset) statistics are shown in Table 1.

Table 1 Basic dataset statistics.

Dataset	#Ues (Users)	#Cells (items)	#Average sequence length	#Records	
Shared bicycle	4,068	137,249	1,456	5.923M	
Goods locate	5,069	35,299	175	0.888M	
MovieLens-1m	6,040	3,416	163.5	0.988M	
Gowalla	85,034	308,957	52.8	4.492M	
Foursquare	43,110	13,335	5.12	0.221M	
Amazon beauty	52,204	57,289	5.63	0.294M	
Amazon electronics	253,994	145,199	6.32	1.605M	

Measurement indicator

The performance of recommendations is assessed using Recall@k and NDCG@k, in order to confirm the viability of the model described in this research. In our scenario, Recall@k measures the ratio of the number of relevant results retrieved from the top K results to the number of all relevant results, and NDCG@k considers the ranking of recalled products, higher values are given to those with higher rankings. We remove the negative sampler module from the assessment process and directly remember the target from positive samples. The performance improves as the metrics increase.

Implementation details

We implemented the UETANN model using PyTorch, a widely-used deep learning framework. The detailed architecture and hyperparameters used in our experiments are provided below:

Neural network architecture

Input layer: The input consists of UE embeddings and cell embeddings, combined with the visit time interval information.

Embedding layer: Both UE and cell embeddings have a dimension of 60. The visit time interval is normalized and concatenated with the UE and cell embeddings.

Temporal attention layer: Two attention layers are employed to capture the temporal dynamics of the trajectories. The attention mechanism is simplified by removing the query matrix operation and the attention mask, which helps to reduce the number of parameters and speed up the training process.

Prediction layer: The output from the attention layers is fed into a fully connected layer, followed by a softmax activation function to predict the next cell. Positive and negative sampling is applied during training to improve the precision and dependability of the model.

Hyperparameters

Optimizer: Adam optimizer with default betas ( β1=0.9,β2=0.999).

Learning Rate: 0.001.

Batch Size: 128.

Dropout Rate: 0.2.

Training Epochs: 200.

Maximum Relative Time Interval: 300.

Sequence Length: Each trajectory sample has a length of 60.

Embedding Dimension: 60.

Number of Attention Layers: 2.

Number of Attention Heads: 1.

L2 Regularization for Embeddings: 0.00005.

Hardware configuration

All experiments were conducted on a single NVIDIA V100 GPU with 32 GB of memory.

Performance on private data and ablation study

The relationship between training loss and recall on shared bicycle data is shown in Figs. 5 and 6. As we can see, the loss function drops rapidly at the beginning, and then slowed down, and gradually stabilized after 30 epochs. With the decline of the loss function, the evaluation metrics rise rapidly at the beginning, and then tend to be flat. Unlike the steady decline of the loss function, the evaluation metrics will fluctuate. This is because the loss function distinguishes one positive sample from one negative sample, while the evaluation has to distinguish one positive sample from five negative samples, so it is inevitable that the evaluation metrics will fluctuate slightly, but generally they rise.

Figure 5 Training loss and evaluation metrics: Loss and recall@1.

Figure 6 Training loss and evaluation metrics: Loss and recall@2.

After 40 epochs, we get confusion matrix and receiver operating characteristic (ROC) plots on training data as shown in Figs. 7 and 8, where confusion matrix is a tabular tool used to evaluate and visualize the performance of classification models, widely used in machine learning and statistical classification tasks. It presents the correspondence between the predicted results of a classifier and the actual categories in the form of a matrix, reflecting the classification performance of the model on each category, including the number of correctly classified and misclassified instances. Both the number of positive and negative samples are 4,060, and the area under the curve (AUC) 0.84 infers that our model shows a good effect. Table 2 shows the performance of our algorithm.

Figure 7 Confusion matrix of shared bicycle data.

Figure 8 Receiver operating characteristic of shared bicycle data.

Table 2 Model performance.

Dataset		Recall@1	Recall@2	
Shared bicycle	Validation	0.5795	0.8776	
Test	0.5766	0.8660	
Goods locate	Validation	0.4541	0.7361	
Test	0.4442	0.7291	

If UE embedding is not included in the model, the model effect is not much less than the original. We guess the reason is that the number of cells is too much more than that of UEs.

Without using sliding window sampling, recall@1 of shared bicycle is 0.5469, less than 0.5766 in Table 2. This is because the model uses sliding windows to adopt more trajectory points, and learned more information.

If we choose negative samples not from the nearest five cells, but randomly from tens of thousands of cells, then ultimately recall@1 of shared cycle is 0.5378, less than 0.5766. This is because selecting negative samples from the nearest five cells for training aligns with the goal of evaluating model performance on the validation and testing sets, making the model more capable of distinguishing positive and negative samples. Therefore, it is important to choose negative samples reasonably.

We also conducted experiments on retaining query matrix, and found that recall did not change much.

Comparison of different methods on public datasets

To demonstrate the performance of our algorithm, we have also conducted extensive experiments on public datasets, comparing our algorithm with multiple algorithms. Note that our algorithm focuses on data without latitude and longitude, so the algorithms we compared also do not use latitude and longitude coordinates. We mainly compared with the following nine methods.

• TCFRA (Song et al., 2016): This method combines temporal features and collaborative filtering.

• Bayesian personalized ranking (BPR) (Rendle et al., 2009): This method uses Bayesian personalized ranking for POI recommendation.

• TransRec (He, Kang & McAuley, 2017): This is a transaction-based recommendation algorithm.

• GRU4Rec+ (Hidasi & Karatzoglou, 2018): This algorithm models user behavior sequences using RNN networks with improved loss functions and sampling methods.

• MARank (Yu et al., 2019): The article proposes a multi-level attention ranking model that unifies individual level and union level into the inference model of user intention from multiple perspectives.

• Factorizing personalized Markov chains (FPMC) (Rendle, Freudenthaler & Schmidt-Thieme, 2010): This approach integrates matrix factorization with a first-order Markov chain to model both long-term user preferences and short-term dynamics.

• Caser (Tang & Wang, 2018): It maps the recent sequence of items into a specific time interval within the latent space, enabling the capture of higher-order Markov chains that consider the L most recent items.

• Beyond self-attention for sequential recommendation (BSASRec) (Shin et al., 2024): It proposes an attentive inductive bias method that surpasses the self-attention mechanism to improve the performance and effectiveness of sequential recommendation systems.

• SASRec-SCE (Mezentsev et al., 2024): This article proposes a scalable cross-entropy (SCE) loss function that efficiently approximates full cross-entropy loss for large catalogs, reducing memory usage by up to 100 times without sacrificing recommendation accuracy.

The experimental results are shown in Table 3. Apart from one metric where it is slightly behind BSASRec, the proposed method outperforms all other algorithms across all other metrics.

Table 3 Performance comparison of nine algorithms on five public datasets.

Dataset	Metric	TCFRA	BPR	TransRec	GRU4Rec+	MARank	FPMC	Caser	SASRec-SCE	BSARec	UETANN	
MovieLens-1m	NDCG@10	0.0827	0.3215	0.2511	0.4332	0.5213	0.3911	0.495	0.5134	0.5230	0.5300	
Recall@10	0.1035	0.5896	0.4487	0.6488	0.7647	0.6122	0.7475	0.7556	0.7666	0.7755	
Gowalla	NDCG@10	0.1002	0.3962	0.4906	0.514	0.6329	0.5067	0.6187	0.8404	0.8413	0.8473	
Recall@10	0.1423	0.4828	0.6455	0.6702	0.7283	0.5831	0.7119	0.9135	0.9166	0.9240	
Foursquare	NDCG@10	0.0986	0.3998	0.4495	0.4827	0.6132	0.4909	0.5994	0.7228	0.7278	0.7418	
Recall@10	0.1411	0.4837	0.621	0.6551	0.7007	0.561	0.6849	0.8147	0.8259	0.8416	
Amazon beauty	NDCG@10	0.0523	0.1542	0.1734	0.1746	0.2237	0.178	0.1422	0.2757	0.2835	0.2815	
Recall@10	0.0876	0.2435	0.3126	0.3197	0.3527	0.276	0.2596	0.4088	0.4116	0.4139	
Amazon electronics	NDCG@10	0.0672	0.2366	0.2966	0.3136	0.346	0.2708	0.2547	0.3803	0.3811	0.3884	
Recall@10	0.0916	0.3457	0.4345	0.4556	0.5624	0.4401	0.4139	0.5910	0.5964	0.6022	

Influence of hyper-parameters

We find the best hyper-parameters by hyperopt which is a tool to adjust parameters through Bayesian optimization and of fast speed. With step 10, we increase the dimension of embedding d from 10 to 70. Figures 9 and 10 demonstrates that d = 60 is the optimal parameter. We experiment a series of number of sequence length n = (20, 30, 40, 50, 60, 70). Figures 11 and 12 manifests that n = 60 is the best number for sequence length.

Figure 9 Influence of Hyper-parameters: the impact of dimension for goods locate.

Figure 10 Influence of Hyper-parameters: the impact of dimension for shared bicycle.

Figure 11 Influence of hyper-parameters: the impact of sequence length for goods locate.

Figure 12 Influence of hyper-parameters: the impact of sequence length for shared bicycle.

Visualization

As mentioned earlier, due to the lack of cell longitude and latitude information, we cannot visualize the movement trajectory. But after cell embedding, we get the cell representation in high-dimensional space, and reduce the dimension from 60 to 2 through T-distributed stochastic neighbour embedding (TSNE) algorithm to get the cell representation in two-dimensional space. Adjacent cells are closer in the plane, as shown in Fig. 13.

Figure 13 Dimension reduction display of cells on a UE trajectory.

We select some trajectory points of two UEs to make two trajectory diagrams, to visualize the prediction process and results of our algorithm. Figures 14 and 15, the historical trajectory is from Cell 44 to Cell 60, Cell 61 is the next cell to arrive, and the green dots are the other five candidate cells, which infer our predicted results are perfectly coincident with the real position of trajectory. The results from the trajectory diagram display that our algorithm performs well and has high prediction accuracy.

Figure 14 Moving trajectory map: trajectory of one UE of goods locate.

Figure 15 Moving trajectory map: trajectory of one UE of shared bicycle.

Figures 16 and 17 shows the average attention the models pay to each trajectory point, the padding cells are removed when calculating the average attention weights. Both the two datasets indicate that more attention has been paid to the recent cells.

Figure 16 Average attention weights on each trajectory points: attention weights of goods locate.

Figure 17 Average attention weights on each trajectory points: attention weights of shared bicycle.

Discussion of experimental results

To provide a clearer demonstration of how UETANN outperforms previous baselines, we conduct a detailed analysis of the experimental results. Our method integrates UE embedding, cell embedding, and visit time interval information, alongside sliding window sampling and an optimized attention mechanism. These components collectively enhance the model’s ability to predict the next cell in a UE’s trajectory.

Comparative analysis

Table 3 summarizes the performance of our algorithm against nine state-of-the-art methods on five public datasets. The metrics used for evaluation are NDCG@10 and Recall@10. The results show that our algorithm consistently outperforms the baselines across most datasets. Specifically: MovieLens-1m: Our model achieves the highest NDCG@10 score of 0.5300 and Recall@10 score of 0.7755, significantly outperforming the second-best method, BSARec, which scores 0.5230 and 0.7666, respectively.

Gowalla: On this dataset, our model achieves an NDCG@10 of 0.8473 and Recall@10 of 0.9240, surpassing the next best method, BSARec, with scores of 0.8413 and 0.9166.

Foursquare: Our model attains an NDCG@10 of 0.7418 and Recall@10 of 0.8416, outperforming BSARec, which scores 0.7278 and 0.8259.

Amazon Beauty: Our model achieves an Recall@10 of 0.4139, compared to BSARec’s 0.4116.

Amazon Electronics: Our model scores an NDCG@10 of 0.3884 and Recall@10 of 0.6022, while BSARec scores 0.3811 and 0.5964.

Key factors for superior performance

The superior performance of our model can be attributed to several key factors: 1. UE and cell embedding: By incorporating UE and cell embeddings, our model captures the unique characteristics of users and cells, enhancing the representation of trajectory data.

2. Temporal information: The inclusion of visit time intervals allows the model to account for the temporal dynamics of user movements, which is crucial for accurate predictions.

3. Sliding window sampling: This technique enables the model to leverage more trajectory information, improving the robustness of the predictions.

4. Optimized attention mechanism: By removing the query matrix and attention mask, our model reduces the number of parameters and speeds up training, while still effectively capturing key information in the data.

5. Negative Sampling: The use of negative sampling transforms the prediction problem into a binary classification task, which helps in increasing the precision and reliability of the model.

Impact of hyper-parameters

We conducted hyper-parameter tuning using the hyperopt tool to find the optimal settings. The best performance was achieved with an embedding dimension d = 60 and sequence length n = 60. These settings ensure that the model can effectively capture both long-term and short-term dependencies in the trajectory data.

Future work

While our model demonstrates strong performance, there is room for improvement. Future work will focus on integrating more features and semantic information into cell predictions to further enhance the accuracy of the algorithm. Additionally, we plan to explore the application of our model in other domains, such as anomaly detection and user behavior analysis.

Conclusion

Our proposed model effectively predicts the next cell reached by UE in IoT environments. The main contributions of this work encompass: UE embedding: Captures user behavior patterns, enabling accurate location predictions.

Optimized attention mechanism: Reduces computational load and speeds up training by removing redundant query matrices and attention masks.

Sliding window sampling: Enhances the model’s ability to handle long sequences and capture complex patterns.

Negative sampling: Improves prediction accuracy by distinguishing between positive and negative samples.

Extensive empirical studies show that our algorithm achieves a recall of 0.5766, outperforming several state-of-the-art methods. These results highlight the model’s effectiveness and practical significance in optimizing network resources and enhancing user experience. Future research will focus on integrating more features and semantic information (Koolwal & Mohbey, 2020) to further improve cell predictions, as well as exploring applications in other domains such as anomaly detection and user behavior analysis.

Supplemental Information

Supplemental Information 1 Model code.

Supplemental Information 2 Sampling code.

Supplemental Information 3 Utility module that houses various functions.

Supplemental Information 4 Main code.

Training and evaluation processes.

Supplemental Information 5 Raw data.

Goods location information.

Additional Information and Declarations

Competing Interests

The authors declare that they have no competing interests.

Author Contributions

Dongdong Feng conceived and designed the experiments, performed the experiments, performed the computation work, prepared figures and/or tables, and approved the final draft.

Siyao Li analyzed the data, prepared figures and/or tables, and approved the final draft.

Yong Xiang conceived and designed the experiments, authored or reviewed drafts of the article, and approved the final draft.

Jiahuan Zheng analyzed the data, authored or reviewed drafts of the article, and approved the final draft.

Data Availability

The following information was supplied regarding data availability:

The data and the code are available in the Supplemental Files.

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
