# Peer review of "A user-embedded temporal attention neural network for IoT trajectories prediction"

_PeerJ Computer Science, doi:10.7717/peerj-cs.2681_

## Round 0.1 · original submission · Major Revisions

Dear Authors,

Thank you for submitting your article. Based on reviewers' comments, your article has not been recommended for publication in its current form. However, we encourage you to address the concerns and criticisms of the reviewer and to resubmit your article once you have updated it accordingly.

Equations should be used with correct equation number. Please do not use “as follows”, “given as”, etc. Explanation of the equations should also be checked. All variables should be written in italic as in the equations. Their definitions and boundaries should be defined. Necessary references should be provided. Many of the equations are part of the related sentences. Attention is needed for correct sentence formation. Furthermore, please pay special attention to the usage of abbreviations. Spell out the full term at its first mention, indicate its abbreviation in parenthesis and use the abbreviation from then on.

Best wishes,

Reviewer 1 ·

Basic reporting

The paper introduces a novel User-Embedded Temporal Attention Neural Network specifically designed for IoT trajectory prediction. One of the main contributions is the UE (User Equipment) embedding and cell embedding, which allow the model to effectively capture the relationships between users and their movement patterns across different cells without relying on GPS coordinate data. Another key innovation is the use of sliding window sampling, which enhances the model’s ability to process longer trajectories, providing a more comprehensive view of user behavior over time. The paper also improves the attention mechanism by removing the query matrix and attention mask, which reduces the number of parameters and speeds up model training, making the model more efficient without sacrificing performance. Furthermore, the model uses negative sampling to optimize training by selecting adjacent cells for more realistic trajectory prediction, improving both precision and speed. In terms of experimentation, the model demonstrates superior performance compared to several state-of-the-art methods on both private and public datasets, achieving high recall and NDCG scores. The paper's findings also offer practical implications for real-time IoT mobility management and anomaly detection, highlighting its potential for real-world applications. Additionally, the study provides valuable insights into embedding-based approaches for sequential recommendation tasks, showing how temporal and positional information can be effectively leveraged for improved trajectory prediction. General speaking, the proposed ideas in this paper are quite interesting and could be considered for publication. Beside good points of this paper, I also have some comments for authors to improve their paper’s quality, including:
1. First of all, for the content in introduction section, it provides a general context for IoT trajectory prediction, but the specific problem being addressed (the limitations of existing models for trajectory prediction) could be made more explicit. Emphasize the key challenges that your model aims to solve right at the beginning to grab the reader’s attention.
2. Moreover, while the introduction touches on some contributions (e.g., UE embedding, attention mechanism optimization), it lacks a clear, concise statement outlining the novelty of your approach compared to existing methods. Adding a bullet-pointed list of contributions at the end of the introduction would help readers quickly grasp the unique aspects of your work.
3. Furthermore, the literature review mentions various models like LSTM, RNN, and attention-based methods but does not thoroughly compare these models with your proposed approach. Provide a more detailed analysis of how each of these methods has limitations (e.g., handling long sequences, reliance on GPS data), and explain how your model overcomes those limitations. The literature review could be more structured by grouping related works into thematic categories (e.g., "Sequential Recommendation Models", "Attention Mechanisms in IoT", "Trajectory Prediction without GPS Data"). This would make it easier for readers to follow the progression of related research and see where your work fits in.
4. Consider adding more subheadings within the introduction and literature review to break down the sections into more digestible parts. For example, in the introduction, you could have subheadings like “Motivation” and “Contributions,” while in the literature review, you could add subheadings like “Existing Approaches for Trajectory Prediction” and “Limitations of Current Methods.” This would improve the overall readability and navigation of the paper.
5. For the methodology of this paper, while the paper introduces the multi-modal embedding layer, attention layer, and prediction layer, the description of how these components interact in the overall architecture is somewhat brief. Including a detailed explanation of each layer’s role and how they contribute to the final prediction would help readers better understand the model’s structure. A more detailed flow diagram could also be added to visualize the model’s architecture.
6. Finally, more discussions on the experimental results to demonstrate clearer on how our proposed technique can outperformance previous baselines.

Experimental design

No comment.

Validity of the findings

Please refer to my basic reporting section.

Additional comments

No comment.

Cite this review as

Reviewer 2 ·

Basic reporting

The paper proposes a user-embedded temporal attention neural network for IoT trajectories prediction. The algorithm combines UE embedding and cell embedding, and uses sliding window sampling to process UE trajectory data. The attention mechanism is used to obtain key information in the data and reduce the number of parameters .Prediction layer uses positive and negative sampling to increase precision and dependability. There are some problems in the paper:
(1) In the introduction, you need to connect the state of the art to your paper goals. Please follow the literature review by a clear and concise state of the art analysis. This should clearly show the knowledge gaps identified and link them to your paper goals.
(2) Please reason both the novelty and the relevance of your paper goals.
(3) In addition, the tools for making pictures are not proper, and the original files are of low quality and less clear.
(4) The references can be updated to include recently published related works, such as:
[1] Augmented Intelligence of Things for Emergency Vehicle Secure Trajectory Prediction and Task Offloading
[2] TRACE: Transformer-based continuous tracking framework using IoT and MCS
[3] Trajectory Simplification and Adaptive Map Matching Algorithm for Electric Bicycle

Experimental design

(1) The paper lacks a thorough evaluation and comparison with existing baselines in the field.
(2) The paper does not provide implementation details that would allow for the replication of the study, such as the specific neural network architecture and the use of hyperparameters.
(3) The paper does not compare the algorithm's performance to other existing IoT trajectory prediction methods.

Validity of the findings

(1) In the conclusions, in addition to summarizing the actions taken and results, please strengthen the explanation of their significance. It is recommended to use quantitative reasoning comparing with appropriate benchmarks,especially those stemming from previous work.
(2) The authors need to be more specific about the contribution of this study, such as stating whether the work fills in some knowledge gaps that previous articles have not addressed.

Cite this review as

---

## Round 0.2 · Minor Revisions

Dear authors,

Feedback from the reviewers is now available for your revised paper. It is still not recommended that your article be published in its current format. However, we strongly recommend that you address the issues raised by Reviewer 2 and resubmit your paper after making the necessary changes.

Best wishes,

Reviewer 1 ·

Basic reporting

I have carefully checked all the revisions as well as authors’ feedback within the latest manuscript version of this paper and confirmed that authors have fulfilled all mine as well as other reviewers’ suggestions. All the problems in the previous paper’s versions have been completely resolved. Thus, I thought this paper can be accepted for publication in this form.

Experimental design

No comment.

Validity of the findings

Please refer to my basic reporting section.

Additional comments

No comment.

Cite this review as

Reviewer 2 ·

Basic reporting

The authors have improved the manuscript according to the reviews. Still, I have the following suggestions:
1. Most figures are not clearly enough, especially the road maps, and the authors should improve them.
2. The paper lacks a thorough evaluation and comparison with existing baselines published recently (for example, in 2023 or 2024) in the field.
3. The related works should be updated to include recently published related works
4. The writing is not good enough, and a proof reading by a native speaker is suggested.
5. The organization and presentation can be improved. For example, pseudo code can be given in a table, and variables should be inputted with professional tools, such as mathtypes. Besides, the same variables should have the same format.

Experimental design

check the “1. Basic reporting”

Validity of the findings

check the “1. Basic reporting”

Cite this review as

---

## Round 0.3 · accepted · Accept

Dear Authors,

Thank you for clearly addressing the reviewers' comments. Your paper now seems ready for publication.

Best wishes,

Reviewer 2 ·

Basic reporting

I have no more suggestions.

Experimental design

I have no more suggestions.

Validity of the findings

I have no more suggestions.

Additional comments

I have no more suggestions.

Cite this review as